# Transferrin-Functionalized Liposomes Enhance MAPT-ASO Transport Across a 3D Blood–Brain Barrier Microvascular Network Model

**DOI:** 10.3390/ijms262311347

**Published:** 2025-11-24

**Authors:** Simon Konig, Xinai Shen, Giuseppe Mantovani, Gerlof Sebastiaan Winkler, Zheying Zhu, Emad Moeendarbary

**Affiliations:** 1Department of Mechanical Engineering, University College London, London WC1E 6BT, UK; simon.konig.21@ucl.ac.uk; 2School of Pharmacy, The University of Nottingham, Nottingham NG7 2RD, UK; 3BioRecode Ltd., London EC4N 7BE, UK

**Keywords:** Alzheimer’s disease, tau protein, antisense oligonucleotides (ASO), blood–brain barrier model, targeted liposomal delivery, neurotherapeutics

## Abstract

Tau pathology is a defining hallmark of Alzheimer’s disease (AD), closely associated with cognitive decline. Antisense oligonucleotides targeting the tau-encoding gene *MAPT* (MAPT-ASO) have shown promise in clinical trials, but their therapeutic potential is limited by poor delivery across the blood–brain barrier (BBB). In this study, we developed transferrin (TF)-functionalized liposomes encapsulating MAPT-ASOs and evaluated their transport across a 3D self-assembled microvascular BBB model composed of human brain microvascular endothelial cells, astrocytes, and pericytes embedded in a fibrin hydrogel. Following confirmation of MAPT-ASO efficacy in reducing tau levels and protecting against glutamate-induced axonal degeneration, we observed significantly enhanced extravascular accumulation and sustained delivery of MAPT-ASOs with TF-functionalized liposomes over 24 h, compared to non-functionalized control liposomes. This study presents a novel delivery strategy for a functionally effective tau-targeting anti-sense oligonucleotide (ASO), potentially enabling systemic delivery rather than intrathecal administration. In addition, this study demonstrates the utility of the 3D in vitro BBB model for screening and optimizing brain delivery of nucleic acid-based therapeutics.

## 1. Introduction

Alzheimer’s disease (AD) is a progressive neurodegenerative disorder and the leading cause of dementia worldwide [1]. Its pathological hallmarks include extracellular aggregation of amyloid-β (Aβ) peptides into plaques and intracellular accumulation of hyperphosphorylated tau in the form of neurofibrillary tangles, both of which disrupt synaptic integrity and neuronal homeostasis, leading to widespread neurodegeneration [2]. Recently, several disease-modifying anti-Aβ monoclonal antibody therapies have entered the clinic; however, significant challenges remain due to their relatively modest effects, the need for serial MRI monitoring to detect potentially severe side effects and the fact that only patients in the early stages of AD can be treated [3,4,5,6,7,8]. Targeting tau has been proposed as a potentially superior therapeutic strategy to amyloid, as the formation of neurofibrillary tangles closely coincides with the onset of dementia symptoms and more accurately predicts patterns of clinical decline [9,10,11]. Various strategies to target tau pathology in AD have previously been explored with multiple approaches in late-stage clinical trials [8,12,13]. Nucleic acid-based therapeutics, such as antisense oligonucleotides (ASOs), provide an opportunity to modify disease progression in AD by downregulating overexpressed genes [14]. ASOs targeting the *MAPT* gene (MAPT-ASOs), which encodes for the tau protein, reduce expression by binding to tau mRNA and recruiting RNase H1, which degrades the transcript and prevents tau protein synthesis [14]. In a Phase 1b clinical study, the MAPT-ASO candidate BIIB080 demonstrated a dose-dependent and sustained reduction in cerebrospinal fluid (CSF) total tau and phosphorylated tau (p-tau) and is currently being further evaluated in a Phase 2 trial, thus highlighting the clinical potential of MAPT-ASOs in treating AD [15,16].

A major obstacle in the development of AD therapies is the limited penetration of drugs into the brain due to the restrictive nature of the blood–brain barrier (BBB) [17]. The BBB is a highly selective interface at the brain microvasculature, composed of brain microvascular endothelial cells (BMECs), astrocytes, pericytes, and extracellular matrix components such as the vascular basement membrane and the intraluminal glycocalyx [18]. BMECs are interconnected by complex tight junctions that strongly limit paracellular transport, forcing most therapeutics to rely on the transcellular pathway for entry into the central nervous system (CNS) [19]. As a result, drug penetration is poor, often necessitating high systemic doses to achieve therapeutic concentrations in the brain [17]. ASOs are no exception, as their relatively large size, strong negative charge, and hydrophilicity prevent passive diffusion across the BBB [20]. Consequently, current clinical trials of MAPT-ASOs employ intrathecal administration, an invasive and repeated procedure associated with uneven brain distribution, procedural risks, and limited patient tolerability [15,16]. To address these limitations and enable systemic delivery, ASOs can be incorporated into lipid nanoparticles such as liposomes, which provide protection against nuclease degradation, improve pharmacokinetic properties, and enhance cellular uptake and endosomal escape [21]. Functionalizing these liposomes with surface ligands that bind to receptors overexpressed at BMECs facilitates BBB crossing through receptor-mediated transcytosis, thus allowing for CNS-specific systemic delivery [22,23,24].

Transferrin receptor 1 (TFR1) is preferentially expressed on the luminal surface of BMECs compared to peripheral endothelium, making it an attractive and relatively specific target for delivering substances across the BBB to the CNS [25]. The physiological role of TFR1 in mediating essential iron transport across the BBB highlights its powerful and active endocytic transport pathway for the delivery of drugs for the treatment of AD. One of the most important features of TFR1 in AD is that its expression levels and functional integrity at the BBB appear to be largely preserved despite AD neuropathology. This is a crucial distinction because many other BBB components, such as tight junctions and efflux transporters, exhibit dysfunction or altered expression in AD, which may complicate the delivery of drugs that rely on their normal function [26]. Given that transferrin (TF) is a widely tolerated endogenous protein and its conjugates are compatible with scalable chemical systems (EDC/NHS: carboxyl-to-amine; maleimide–thiol; click: fast azide–alkyne), which facilitate its conjugation to various carriers and reduces immunogenicity, in order to allow TF to compete with abundant plasma transferrin and avoid random lysine conjugation interfering with iron binding [27], TF can be placed on a flexible PEG tether, and the surface density of TF can be manipulated to achieve the required affinity for entering endothelial cells.

To evaluate ASO drug delivery strategies, animal models can be used [21]. However, due to species-specific differences in BBB properties, such as tight junction complexity and TFR1 density, distribution, and ligand-binding characteristics, the translatability of these studies is limited [28]. In vitro BBB models provide an alternative, yet conventional transwell systems fail to capture the 3D architecture, lack dynamic flow and are further biased by introducing a stiff membrane with non-physiological properties at a critical location in the BBB model [29]. Self-assembled BBB organoids offer potential high-throughput tools for permeability studies, including nanoparticle evaluation, but they lack a full BBB architecture, allow only for semi-quantitative assessments and may be complex and lengthy to establish [30,31]. Hydrogel-based approaches present significant advantages, as cells can organize with cell–cell and cell–extracellular matrix (ECM) interactions that reflect in vivo BBB anatomy. Previous studies have evaluated nanoparticles using hydrogel-based transwell-like systems or self-assembled capillary structures in wells with perfusable openings [32,33]. However, these systems still lack fluid flow, an essential factor in BBB physiology and nanoparticle attachment. Microfluidic BBB chips address this limitation by incorporating micro-sized channels that allow controlled fluid flow and simultaneous cell culture [29,34,35,36,37]. Previously, a perfusable BBB microvasculature network (MVN) model has been developed through self-assembly of BMECs, astrocytes, and pericytes within a fibrin hydrogel in a microfluidic chip [38,39]. This model accurately recapitulates in vivo BBB architecture, including the formation of luminal microvessels with astrocytes and pericytes actively aligning and interacting with the endothelium [38,39]. Moreover, physiologically relevant expression and localization of BBB membrane transporters, tight junction proteins, and extracellular matrix components, as well as cytokine production, were observed, with permeability values comparable to those observed in rat brains [38,39]. Importantly, TFR1 expression in BMECs increases during the formation of the MVNs [39] and is ultimately higher in BMECs cultured as MVNs than in 2D monocultures [38], highlighting how such approaches enhance physiological relevance and are therefore particularly suitable for studying TF-mediated delivery. Previous studies have shown that nanoparticle size and functionalization can be effectively evaluated for BBB permeability using this BBB MVN model alone or combined with a tumour spheroid [40,41].

In this study, we investigated whether TF-functionalized liposomes encapsulating MAPT-ASOs could enhance BBB permeability using the previously published and well-characterized 3D BBB MVN in vitro model [38,39,40]. We hypothesized that TF functionalization would enhance the permeability of MAPT-ASO-loaded liposomes across the BBB MVN in vitro model. To test this, TF-functionalized and control liposomes were perfused through the BBB MVNs, and MAPT-ASO delivery was monitored by fluorescent labelling and confocal microscopy to assess translocation across the BBB over 24 h. The results demonstrated that transferrin-functionalized liposomes enabled enhanced transport of the functionally effective MAPT-ASO to extravascular regions within the BBB MVN. These findings reveal a new strategy for MAPT-ASO brain delivery and provide novel methodologies for leveraging the 3D BBB MVN model as a versatile platform for nanoparticle evaluation.

## 2. Results

### 2.1. MAPT-ASO Protects GA-Induced Axonal Degeneration

The efficacy of MAPT-ASO was assessed in SH-SY5Y cells and neurons. After 72 h treatment, ELISA illustrated that MAPT-ASO reduced the pTau181/total ratio compared with vehicle and scrambled ASO (Figure 1A, Appendix A Figure A1). We then characterized the DL-glyceraldehyde (GA) cytotoxicity. At a concentration of 0.7 mM, GA reduced cell viability by approximately 20%. This concentration was chosen for subsequent assays (Figure 1B). An increase in the pTau181/total tau ratio was observed after 0.7 mM GA treatment relative to untreated controls (Figure 1C). Based on these results, 0.7 mM GA was used to induce RA-differentiated neurons and to measure whether MAPT-ASO provides structural protection. CLSM revealed that tubulin was disrupted in neurons treated with GA, resulting in a more than two-fold reduction in average axon length. Neurons treated with MAPT-ASO preserved axon length (Figure 1D,E). Data are mean±SD, *n* = 3. Statistical analysis was performed using one-way ANOVA followed by Dunnett’s test (** *p* < 0.01; **** *p* < 0.0001).

### 2.2. Development of a Transferrin-Conjugated Liposomal Delivery System for MAPT-Targeted Antisense Oligonucleotides

We designed transferrin-functionalized liposomes to promote transferrin-receptor engagement at the blood–brain barrier, with non-functionalized vesicles as the low-transport control (Figure 2A). Liposomes were first prepared by the thin-film hydration method [42]. Then, TF was conjugated to the surface of liposomes at DSPE-PEG(2000)-COOH to TF at a molar ratio of 1:2 to prepare TF-functionalized liposomes (Figure 2B). The size and size distribution of liposomes were measured by dynamic light scattering. The representative size distribution, polydispersity indexes (PDIs) and ζ-potential of the liposomes and TF–liposome preparations are shown in Figure 2C. The encapsulation efficiency of ASO was 48.3 ± 1.6%. TF grafting rate was 39.2 ± 11.2%. Cryo-TEM confirmed predominantly spherical shape and diameters consistent with DLS (Figure 2D,E). In stability studies, both formulations maintained hydrodynamic diameter and PDI with only minor fluctuations across the first 7 days, indicating good colloidal stability (Figure 2F). To further evaluate the liposomes under biologically relevant conditions, the size and ζ-potential of liposomes and TF–liposomes were monitored over 24 h at 37 °C in serum with PBS. Both liposomes and TF–liposomes illustrated a gradual increase in hydrodynamic diameter over time. However, no abrupt enlargement or aggregation was observed (Figure 2G). The particle size remained below ~200 nm throughout the 24 h incubation, indicating acceptable stability under physiological temperature and serum exposure. Similarly, the ζ-potential exhibited moderate fluctuations but remained within a narrow negative range without sudden charge reversal (Figure 2H). The absence of drastic changes in surface charge further supports that neither liposome underwent significant protein-induced aggregation. Collectively, these findings demonstrate that the liposomes and TF–liposomes maintain good short-term stability under physiologically relevant conditions.

### 2.3. Transferrin Functionalization of Liposomes Enhances Cy3-Labelled MAPT-ASO Delivery Across a 3D Microvascular BBB Model

To evaluate whether TF functionalization improves the delivery of Cy3-labelled MAPT-ASO (Cy3-MAPT-ASO) across the BBB, we utilized a previously developed microfluidic in vitro BBB model that closely recapitulates the 3D architecture of human microvasculature. This model has been extensively characterized for the expression of BBB-specific markers [38,39] and has previously been applied to study polymer nanoparticles [40,41].

Following successful assembly of the BBB microvasculature from human BMECs, astrocytes, and pericytes (Figure 3A,D), liposomes were perfused through the microvasculature on day 6. The distribution of Cy3-MAPT-ASOs was assessed by confocal imaging at 30 min and 24 h post-perfusion (Figure 3B). To mimic blood flow-mediated transport within the microvasculature, transient flow of liposome solutions was initiated across the BMEC monolayer and within the established microvascular network (Figure 3C, step 3). The homogenous liposome solution entered the microvasculature exclusively via the media channel without diffusing into the hydrogel, as evidenced by the largely Cy3-negative extravascular spaces immediately after perfusion compared to the strong Cy3 signal within the media channel and microvasculature (Figure 3E). Notably, initial inspection of Cy3-MAPT-ASO signal within the microvasculature revealed that, after 24 h, a more visible Cy3 signal remained in vessels when delivered via non-functionalized liposomes compared to TF-functionalized liposomes (Figure 3G, inset).

Following these observed differences in liposome distribution, we quantified Cy3 fluorescence across defined regions of interest (ROIs) by segmenting signal inside and outside the BBB microvasculature lumens using image processing (Figure 4A) [43].

Quantification of Cy3-MAPT-ASO fluorescence intensity revealed that, 30 min after perfusion, TF-functionalized liposomes showed slightly lower signal than non-functionalized liposomes both within and outside the lumens (Figure 4B). By 24 h, however, TF-functionalized liposomes exhibited markedly higher signal than non-functionalized liposomes, with the difference in signal significantly increased compared to the 30 min timepoint (*p* = 0.002). Post hoc analysis indicated that this increase after 24 h was specific to extravascular regions, where the difference in signal between TF-functionalized and non-functionalized liposomes was significantly greater than both the intraluminal (*p* = 0.017) and the extravascular difference at 30 min (*p* = 0.008).

To further evaluate Cy3-MAPT-ASO delivery beyond the microvasculature, we quantified fluorescence intensity up to 100 μm from the vessel lumen and expressed it as a percentage of intraluminal signal (Figure 4C,D, Appendix A Figure A5). At 30 min, mean permeability values within the 100 μm from the vessel lumen were 89.0% for TF-functionalized and 87.0% for non-functionalized liposomes. Cy3-MAPT-ASO signal intensity decreased gradually with distance from the microvasculature for both groups, as indicated by the negative slope of the fitted trendlines. After 24 h, permeability remained high for TF-functionalized liposomes (82.6% average across 100 μm) but decreased substantially for non-functionalized liposomes (69.5% average across 100 μm). This was further reflected by a 3.6-fold steeper negative slope in the trendlines for non-functionalized liposomes compared to TF-functionalized liposomes, indicating even lower concentration of Cy3-MAPT-ASO further away from the lumen. It is important to note that MVNs of comparable size were analyzed for both liposome groups (Appendix A Figure A4), with lumen diameters only increasing significantly between the 30 min and 24 h timepoints, which is an expected change given the dynamic nature of the BBB model as well as the introduction of fluid flow during liposome perfusion (mean difference of 15 μm, *p* = 0.024).

Overall, these findings indicate that transferrin functionalization enhances the long-term extravascular delivery and retention of Cy3-MAPT-ASO beyond the BBB microvasculature. While early distribution (30 min) was comparable between groups, TF-functionalized liposomes demonstrated significantly greater extravascular accumulation and sustained permeability after 24 h compared to non-functionalized liposomes, suggesting improved transport and tissue penetration over time.

## 3. Discussion

In this study, we evaluated whether transferrin-functionalized liposomes loaded with antisense oligonucleotides targeting the *MAPT* gene (MAPT-ASO) could enhance delivery across the BBB in a 3D in vitro model of human brain microvasculature. After demonstrating that the here-used MAPT-ASO was indeed effective in lowering tau levels and protecting against axonal degeneration, we then showed that transferrin-functionalized liposomes were able to enhance the delivery of the MAPT-ASO across the BBB after 24 h compared to the non-functionalized liposomes. Therefore, this study presents a novel delivery strategy for a functionally effective tau-targeting ASO, which could potentially enable systemic delivery of these promising therapeutics for Alzheimer’s disease. Furthermore, it highlights the utility of the 3D BBB microvasculature network in vitro model as a versatile and translational platform for evaluating the brain delivery of nucleic acid-based therapeutics.

The ASO used here is a fully phosphorothioate oligo directly targeting human MAPT pre-mRNA. By hybridizing to MAPT mRNA, the ASO recruits RNase H1 to cleave the target transcripts, thereby lowering MAPT mRNA and reducing tau protein synthesis [44]. Using a well-established GA-induced axonal degeneration approach in SH-SY5Y–derived neurons [45], we found that GA reduced viability in a dose-dependent manner (0.7–2.8 mM). CLSM revealed marked neurite fragmentation after 0.7 mM GA, which was rescued by co-treatment with the MAPT-ASO. GA significantly shortened axons relative to untreated neurons, whereas GA + MAPT-ASO significantly increased axon length compared with GA alone. In parallel, the MAPT-ASO significantly lowered the pT181/total Tau ratio versus both untreated and scramble-ASO controls. These results illustrate that MAPT knockdown reduces pathogenic tau phosphorylation at Thr181 and mitigates axonal damage under metabolic stress in this neuronal model. These findings align with prior human data showing that intrathecal administration of this MAPT-targeting ASO produces reductions in CSF tau species and engages its intended RNase-H mechanism in patients with early AD [16]. Future studies should evaluate the long-term effects of MAPT-ASO treatment on neuronal connectivity and synaptic function, as well as its efficacy in in vivo models of Alzheimer’s disease.

The liposomes prepared in this study exhibited a particle size of 132.8 nm, which increased only modestly to 139.4 nm after TF functionalization. This size falls within the optimal range for crossing the BBB via receptor-mediated transcytosis and is also favourable for efficient endocytosis by neuronal cells. The slight decrease in PDI from 0.201 to 0.189 further confirms that the liposome population remained highly monodisperse and uniform during TF attachment. The slightly negative ζ-potential (−4.04 and −4.65 mV) indicates minimal electrostatic stabilization; however, steric hindrance from PEG chains likely ensures colloidal stability, while near-neutral charge may reduce nonspecific interactions and improve biocompatibility. During perfusion in the microvasculature BBB network model, some aggregation was visible, further highlighting the need for additional improvements in colloidal stability in future formulations. However, during storage at 4 °C, the liposomes remained stable over 7 days, showing no significant change in size, confirming the robustness of the formulation. Transferrin was covalently conjugated to the liposome surface via the carboxyl group of DSPE-PEG(2000)-COOH, allowing stable grafting while preserving the protein’s orientation and functionality for receptor recognition, which is critical for efficient receptor-mediated uptake. The MAPT-ASO encapsulation efficiency (48.3 ± 1.6%) and TF grafting rate (39.2 ± 11.2%) are sufficient to demonstrate payload delivery and receptor-mediated targeting in this study, though affinity-based or density-based separation techniques could further improve payload retention and reduce variability. Overall, these physicochemical characteristics, combined with TF functionalization, support receptor-mediated transport across the BBB and effective MAPT-ASO delivery to neuronal cells.

As expected, TF-functionalized liposomes improved MAPT-ASO delivery across the BBB within the microvascular network model. While transferrin receptor-mediated delivery has been previously explored, this study advances the field by integrating TF-functionalized liposomes with a perfusable 3D in vitro BBB microvascular platform that recapitulates key anatomical and physiological features of the human neurovascular unit. Unlike conventional 2D monolayer models, this system enables dynamic assessment of ASO transport under flow conditions, provides spatial resolution of liposomal distribution across vascular and extravascular compartments, and offers a more predictive representation of human BBB behaviour due to the use of human cells that interact in a physiologically relevant manner. This combination of targeted nanocarrier design and a biomimetic microvascular model offers a novel and translationally relevant strategy for preclinical evaluation of CNS-directed therapeutics, bridging the gap between in vitro screening and in vivo application. In contrast to prior work using a 3D microvasculature BBB model with iPSC-derived endothelial cells (iPSC-ECs), we employed human primary brain microvascular endothelial cells (BMECs) to evaluate nanoparticle transport. In 2D cultures, BBB-relevant genes, including genes of interest for drug delivery such as TFR1 and Low-Density Lipoprotein Receptor-Related Protein 1 (LRP1), were higher in BMECs compared to iPSC-ECs and human umbilical vein endothelial cells (HUVECs) [40]. However, the difference between iPSC-ECs and BMECs diminished when cultured in the 3D co-culture model, suggesting that interactions with astrocytes and pericytes may play a more critical role than the endothelial cell source, a notion we similarly observed recently, also in HUVECs [38,46]. However, BMECs can sustain microvascular networks for longer periods than iPSC-ECs and, in our experience, are easier to incorporate into the microvasculature model [38].

Earlier studies assessed liposome permeability based on changes in fluorescence intensity within the first 5–30 min after nanoparticle administration, a strategy commonly used for barrier integrity testing with tracers such as dextran. However, while this approach is appropriate for detecting paracellular leakiness of small tracer molecules, TF-functionalized liposomes, being substantially larger in size, primarily cross the BBB via receptor-mediated transcytosis [47]. This process involves luminal binding and endocytic internalization, followed by vesicular trafficking through endosomal compartments and subsequent abluminal fusion to release the cargo into the perivascular space, a sequence of events that involves complex kinetics, which are likely very different from paracellular transport [48,49]. Moreover, PEGylated liposomes in humans (e.g., Doxil^®^) exhibit a circulation half-life of approximately 20 to 50 h, further supporting that longer assessment periods are necessary to capture this sustained exposure. Indeed, we observed a marked increase in MAPT-ASO delivery efficiency across the extravascular space only after 24 h. Within the immediate perivascular region, TF-functionalized liposomes provided only a modest improvement after 30 min, but this effect became much more pronounced by 24 h. This finding is particularly relevant as neurons in vivo are typically located within 10–20 µm of brain capillaries, underscoring that TF-functionalized liposomes can substantially enhance MAPT-ASO delivery to their potential neuronal targets over time.

The hereby observed gradual increase in extravascular accumulation of the labelled-MAPT-ASO is likely the result of rate-limiting processes during receptor-mediated transcytosis. For example, limited availability and oversaturation of TFR1s may initially restrict the number of liposomes that can be transported across the endothelium. However, as TFR1s are actively recycled at the BBB [50], continuous receptor-mediated transcytosis could occur over the 24 h period, leading to the observed gradual increase in MAPT-ASO accumulation within the extravascular space. Other factors, such as exocytosis, which has been proposed as the rate-limiting process during receptor-mediated transport of anti-TFR1 monoclonal antibodies, could also be important [49]. Future studies could employ this platform to further investigate and optimize liposomal and cargo trafficking kinetics, as well as TFR1 recycling dynamics, which have previously been shown to be strongly influenced by ligand–receptor binding affinity [50].

Although liposomes consistently enter cells and exhibit enhanced BBB permeability in in vitro models, studies have demonstrated that, when nanoparticles are handled in complex environments, they are rapidly covered by a layer of adsorbed plasma proteins or interact with other proteins in the culture medium and form protein coronas [51]. For transferrin-targeted formulations, several studies have shown that corona formation can sterically shield surface-bound TF and reduce its effective affinity for the transferrin receptor, leading to diminished receptor-mediated uptake and attenuated brain delivery [52]. In this study, we performed preliminary stability assessments in a PBS and FBS mixture. Both liposomes and TF–liposomes maintained relatively stable particle sizes and ζ potential at 37 °C in serum, indicating that the formulations remain stable under simplified physiological conditions. Nevertheless, such in vitro assays cannot fully reproduce the dynamic and compositionally diverse protein corona that forms in vivo, and therefore do not fully predict the extent to which TF will remain functionally accessible in the bloodstream. Future work may focus on determining how its composition influences TF receptor binding and subsequent biodistribution. In addition, formulation strategies such as adjusting PEG architecture or incorporating surface chemistries that mitigate ligand masking will be explored to better preserve the specificity of transferrin–receptor interactions after systemic administration [53].

Since liposomes circulate through the brain microvasculature under flow, which likely affects margination and ligand-receptor binding [54,55], we applied transient fluid flow using a simple hydrostatic pressure-driven setup, as previously described [56,57,58]. This approach produced a more homogeneous distribution of liposomes across the MVNs compared to static conditions. Consistent with prior reports, we observed that transient fluid flow also promoted larger vessel diameters [57]. Due to the induced shear stress and the associated beneficial signalling in BMECs [59], previous studies have shown that fluid flow can extend BBB MVN model viability and enhance MVN perfusability [56,57,60]. Therefore, future studies incorporating stable flow generated by micropumps could support long-term liposome testing, including pre-conditioning and repeated dosing. Integrating such flow systems with robotic handling and automated image analysis would convert the BBB MVN model into a predictive high-throughput platform, as exemplified by other commercially available BBB organ-on-chip systems [34]. The self-assembled BBB MVN model recapitulates essential microvascular architecture and employs human-derived cell types, features that are expected to enhance translational fidelity relative to simpler BBB in vitro models. Future studies should pursue rigorous validation and directly benchmark the model’s predictive accuracy against in vivo and human clinical data, thereby establishing its suitability as a high-throughput platform for systematic evaluation of BBB drug delivery strategies [29].

Our study is limited by its focus on liposome-mediated MAPT-ASO delivery in healthy conditions as well as the lack of in vivo investigations. Given that BBB dysfunction is a key feature of AD pathology, future work should evaluate liposome-mediated MAPT-ASO delivery in an AD context. Previous studies have modelled AD in BBB MVNs by co-culture with neurons carrying familial AD (FAD) mutations, resulting in significantly altered gene expression and permeability in MVNs [56,58]. However, MVN morphology and perfusability were also significantly altered, making efficient nanoparticle perfusion in AD-impacted MVNs difficult to achieve. Future in vivo studies should focus on evaluating whether intravenously administered TF-functionalized liposomes enhance the transport of MAPT-ASO into the brain parenchyma and whether this results in effective reduction in tau mRNA, total tau, and phosphorylated tau levels. Using humanized transgenic animal models, for example, TfR^mu/hu^ knock-in (KI) mice, which express a humanized transferrin receptor 1 at the BBB, and human tau^tg/−^ mice, which express the human MAPT gene on a murine tau null background, allows simultaneous assessment of receptor-mediated BBB transport and target engagement of MAPT-directed therapeutics under human-relevant conditions [61,62,63,64]. In combination with quantitative methods to assess MAPT-ASO penetration across the BBB, such as the capillary depletion technique, fluorescent labelling with co-localization analysis, and blocking or competition studies to confirm transferrin receptor dependence, these experiments would provide comprehensive validation of the liposomal delivery strategy presented here. Given the enhanced biomimicry of the 3D BBB model used in this study, the presented results are expected to have improved translatability to in vivo systems.

## 4. Materials and Methods

### 4.1. Study Design

This study was designed to (1). evaluate the effectiveness of a MAPT-ASO in reducing tau levels and protecting against GA-induced axonal degeneration and (2). assess the ability of TF-functionalized liposomes to deliver the MAPT-ASO across a 3D in vitro BBB model. To assess MAPT-ASO efficacy, pT181/total tau ratios were quantified following treatment with MAPT-ASO, scrambled ASO, or vehicle-only controls, while protection against GA-induced axonal degeneration was evaluated relative to untreated samples. For BBB transport experiments, Cy3-MAPT-ASO-loaded liposomes with or without TF-functionalization were compared. Cy3-MAPT-ASO delivery was assessed at two timepoints: 30 min and 24 h post-perfusion through the BBB model.

### 4.2. Materials

Cells: SH-SY5Y cells (Manassas, ATCC, VA, USA), green fluorescent protein (GFP)-labelled human BMECs (AngioProteomie, Boston, MA, USA; #cAP-0002GFP, Lot: 2022062302), human brain vascular pericytes (ScienCell, Carlsbad, CA, USA; Cat#1200, Lot: 33155), and human brain astrocytes (ScienCell, #1800, Lot: 34727).

Cell culture media: SH-SY5Y media (Essential Medium Eagle Medium (EMEM, Sigma Life Science, St. Louis, MO, USA), 10% (*v*/*v*) fetal bovine serum (FBS, Sigma Life Science), 1% (*v*/*v*) Penicillin–Streptomycin (P/S, Sigma Life Science)), BMEC media (VascuLife VEGF medium (Lifeline Cell Technology, Frederick, MD, USA), 10% FBS (Life Technologies, Carlsbad, CA, USA), pericyte medium (ScienCell), and astrocyte medium (ScienCell).

Other reagents for cell culture: 0.25% trypsin-EDTA (Gibco, Paisley, UK), Phosphate-buffered saline (PBS, Gibco), human fibronectin (Life Technologies), TrypLE Express (Life Technologies), and 0.05% trypsin-EDTA (ScienCell).

ASOs: Scrambled ASO (sequence: 5′-ACCCTTCGAGCTTGAGAGTT-3′) or MAPT-ASO (sequence: 5′-GCTTTTACTGACCATGCGAG-3′), unlabelled or labelled with Cyanine3 (Cyanine3) (Genscript, Oxford, UK). Synthesized with a phosphorothioate backbone.

Liposome preparation: DSPC, CHOL, DSPE-PEG(2000) and DSPE-PEG(2000)-COOH (Avanti Polar Lipids, Alabaster, AL, USA), chloroform (Thermo Fisher, Paisley, UK), 1-Ethyl-3-(3-dimethylaminopropyl)carbodiimide hydrochloride (EDC·HCl) and N-hydroxysulfosuccinimide (sulfo-NHS, Thermo Fisher), transferrin (TF, Sigma, Life Science), 300,000 MWCO Spectra/Por^®^ Dialysis Membrane (Cole-Parmer, St. Neots, UK), 10,000 molecular weight cut-off (MWCO) SnakeSkin™ Dialysis Tubing (Thermo Fisher), Zetasizer Nano (Malvern Instruments Ltd., Malvern, UK).

Microfluidic BBB in vitro model: Polydimethylsiloxane (PDMS, Sylgard 184 Silicone Elastomer Kit Dow, Dow, Cheadle, UK), #1 glass coverslips (Agar Scientific, Stansted, UK), bovine fibrinogen (Merck Life Science, Gillingham, UK), PBS (calcium/magnesium free, Life Technologies), thrombin (Sigma-Aldrich, St. Louis, MO, USA), BBB model culture medium (Vasculife VEGF endothelial medium (Lifeline Cell Technology) prepared with 0.1875 U/mL of heparin and 2% FBS).

Immunofluorescence staining: Paraformaldehyde (PFA; Electron Microscopy Sciences, Hatfield, PA, USA), BSA (Sigma-Aldrich), Triton X-100 (Sigma-Aldrich), normal goat serum (Abcam, Cambridge, UK), Alexa Fluor^®^ 488 anti-beta III Tubulin antibody (#ab195879, Abcam), anti-glial fibrillary acidic protein antibody (#ab33922, Abcam), anti-rabbit Alexa Fluor 647 (#4414, Cell Signaling, Danvers, MA, USA), Phalloidin labelling probe (#A12381, Invitrogen, Paisley, UK), and DAPI (Sigma-Aldrich).

Other reagents: Lipofectamine 2000 (Thermo Fisher, UK), Cell lysis buffer (Radio-Immunoprecipitation Assay buffer (RIPA, Thermo Fisher), phosphatase and protease inhibitor (Thermo Fisher)), pT181 tau and total tau ELISA assay kits (Thermo Fisher), glyceraldehyde (GA, Thermo Fisher), 3-(4,5-Dimethylthiazol-2-yl)-2,5-Diphenyltetrazolium Bromide (MTT, Thermo Fisher), and dimethyl sulfoxide (DMSO, Thermo Fisher).

### 4.3. Cell Culture, ASO Transfection and Protein Extraction

SH-SY5Y cells (ATCC, VA, USA) were cultured in Essential Medium Eagle Medium (EMEM, Sigma Life Science) supplemented with 10% (*v*/*v*) fetal bovine serum (FBS, Sigma Life Science) and 1% (*v*/*v*) Penicillin–Streptomycin (P/S, Sigma Life Science) in 6-well plates for 24 h. Cells were then transfected with 1.35 mM of scrambled ASO or ASO using Lipofectamine 2000 (Thermo Fisher). Scrambled ASO sequence: 5′-ACCCTTCGAGCTTGAGAGTT-3′; ASO sequence: 5′-GCTTTTACTGACCATGCGAG-3′. After an additional 72 h of incubation, cell lysates were prepared. First, SH-SY5Y cells were detached by 0.25% trypsin-EDTA (Gibco) for 2 min. After the cells were centrifuged at 125× *g* for 5 min, the resulting pellets were washed twice with ice-cold PBS (Gibco). Radio-Immunoprecipitation Assay buffer (RIPA, Thermo Fisher) supplemented with 1× phosphatase and protease inhibitor (Thermo Fisher) was prepared as the lysis buffer to lyse cell pellets for 30 min on ice. Then, the lysates were centrifuged at 12,500 rpm for 15 min at 4 °C. Subsequently, the supernatant was collected and transferred into a new microcentrifuge tube for subsequent use.

### 4.4. Enzyme-Linked Immunosorbent (ELISA) Assay

ELISA assays for pT181 and total tau (Thermo Fisher) were performed according to the manufacturer’s protocols. For pT181 quantification, 50 µL of each sample was mixed with 50 µL of detection antibody solution and incubated at 4 °C for 17 h. For total tau quantification, 50 µL of each sample was mixed with 50 µL of standard dilution buffer and incubated at room temperature for 2 h. Following incubation, wells were aspirated and washed four times with 1× Wash Buffer. Subsequently, 100 µL of the corresponding conjugate solution (anti-rabbit IgG HRP for pT181; human total tau–biotin conjugate followed by streptavidin–HRP for total tau) was added to each well and incubated for 30 min at room temperature. After washing, 100 µL of stabilized chromogen was added and incubated for 30 min in the dark. Reactions were terminated with 100 µL of Stop Solution, and absorbance was measured at 450 nm within 2 h.

### 4.5. Cell Viability Assay

SH-SY5Y cells were cultured in minimum essential medium supplemented with 10% *v/v* FBS and 1% *v/v* P/S in 96-well plates for 24 h, followed by treatment with glyceraldehyde (GA, Thermo Fisher) at a concentration of 0, 0.7, 1.4, and 2.8 mM. All assays were performed after an additional 24 h of incubation. MTT solution was then added to each well to a final concentration of 0.455 mg/mL. After 2 h of incubation, the 3-(4,5-Dimethylthiazol-2-yl)-2,5-Diphenyltetrazolium Bromide (MTT, Thermo Fisher) solution was removed, and 100 µL of dimethyl sulfoxide (DMSO, Thermo Fisher) was added to each well. The absorbance was measured at 490 nm by a Tecan plate reader. Data was analyzed using One-way analysis of variance (ANOVA).

### 4.6. Neuron Differentiation

Based on slight modifications of methods previously reported by Uras et al. [45], SH-SY5Y cells were differentiated using the established three-stage protocol and then challenged with GA to generate the GA-induced neuronal model. Specifically, SH-SY5Y cells were diluted to 1.5 × 10^5^ cells/mL, and 2 mL of the suspension was seeded into 35 mm Petri dishes. After 2 days, the medium was replaced with differentiation medium #1 and subsequently changed every 48 h for 5 days. After that, the cells were cultured in differentiation medium #2 for 2 days. Cells were then detached and seeded onto extracellular matrix (ECM) coated dishes. For ECM coating, the ECM solution was diluted 1:100 in cold DMEM, and 2 mL of the mixture was added to each 35 mm dish. After overnight incubation, the solution was aspirated and the dishes were air-dried. One day after seeding on ECM-coated dishes, the medium was replaced with differentiation medium #3 and then changed every 3 days to maintain neuronal health. Neurons were treated either with 0.7 mM GA alone or with GA plus 100 nM of ASO delivered using Lipofectamine 2000 and incubated for 72 h.

### 4.7. Two-Dimensional Immunofluorescence Staining, Imaging, and Neurite Analysis

Glass coverslips were coated with poly-L-lysine for 1 h at room temperature, rinsed three times with sterile water, and then dried completely. SH-SY5Y cells were seeded on a glass-based dish for differentiation. Before fixation, cells were rinsed briefly in PBS-T and were fixed in 4% PFA for 10 min at room temperature. Samples were washed three times with PBS-T and blocked with 1% BSA for 30 min. After blocking, samples were incubated overnight at 4 °C with Alexa Fluor^®^ 488 anti-beta III Tubulin antibody (#ab195879, Abcam) and washed three times in PBS-T. Samples were subsequently stained with DAPI for 15 min and then washed three times in PBS-T. The samples were imaged by Axio Observer Z1/7 Confocal Microscope. Z-stack images were acquired covering a distance of 138 μm. NeuronJ plug-in was performed after background subtraction, with distinct colours assigned to individual traces. Neurite lengths were quantified with the Measure tracing function, and results were analyzed by ANOVA.

### 4.8. Liposome Preparation

The liposomes were prepared by the thin-film hydration method. DSPC, CHOL, DSPE-PEG(2000) and DSPE-PEG(2000)-COOH (Avanti Polar Lipids) were dissolved in chloroform (Thermo Fisher) with a molar ratio of 2:1:0.11:0.021; then, the solution was evaporated at 35 °C under a vacuum condition to remove the organic solvent until the thin film formed. Residual chloroform was further removed by overnight incubation in a vacuum desiccator at room temperature. The thin film was then hydrated with 4 mL of PBS with Cyanine3-MAPT-ASO (Cy3-MAPT-ASO). The suspension was vortexed and sonicated (40 kHz, 25 °C, Allendale Ultrasonics, Hoddesdon, UK) at room temperature for 10 min to prepare liposomes containing ASO and followed by extrusion 21 times through a 100 nm polycarbonate membrane to reduce particle size. Free Cy3-MAPT-ASOs were separated from liposomes by dialysis through 10,000 molecular weight cut-off (MWCO) SnakeSkin™ Dialysis Tubing (Thermo Fisher). After the MAPT-ASO-loaded liposome was prepared, it was activated by 1-Ethyl-3-(3-dimethylaminopropyl)carbodiimide hydrochloride (EDC·HCl) and N-hydroxysulfosuccinimide (sulfo-NHS, Thermo Fisher) for 15 min at room temperature at pH 5.5. TF was added to the active liposome solution and incubated overnight at 4 °C at pH 7.5. Free TF was separated from TF-ASO–liposome by dialysis through 300,000 MWCO Spectra/Por^®^ Dialysis Membrane (Cole-Parmer).

### 4.9. Particle Size Determination and Morphology Observation of Liposomes

The Zetasizer Nano (Malvern Instruments Ltd.) was used to characterize the particle size and ζ-potential of the liposomes. The morphology of the liposome was determined by cryogenic transmission electron microscopy (JEM 2100+, JEOL, Tokyo, Japan). Liposomes and TF–liposomes (n = 3) were resuspended in PBS, PBS-FBS (1:1, *v*/*v*) to mimic the condition in biological fluids and examine the stability for 7 days and 24 h. The variations in the average diameter in these suspensions were determined by DLS.

### 4.10. Determination of Encapsulation Efficiency and Conjugation of Transferrin

The fluorescence intensity of Cy3-MAPT-ASO was used to prepare a calibration curve. After dialysis, the fluorescence intensity of samples was determined via Plate Reader (Tecan, Männedorf, Switzerland), the Cy3 excitation spectrum was obtained at 585 nm emission, and the emission spectrum was obtained at 540 nm excitation, gain was 100. The encapsulation efficiency, expressed as entrapment percentage, was calculated through the following relationship (1):(1)$$Encapsulation\;efficiency\;\%=(C_{A}÷C_{AT})\times 100$$
where C_A_ is the MAPT-ASO concentration in dialysed liposomes, and C_At_ is the MAPT-ASO concentration initially introduced.

Total TF (Thermo Fisher) input was normalized to 100%. A TF calibration curve was generated from serial dilutions of TF. Each standard was mixed 1:1 with liposomes and processed identically to samples. Liposomes and TF served as control groups. Equal volumes were resolved by SDS–PAGE on the same gel, followed by Coomassie staining. Band intensity of TF was quantified in ImageJ (version 1.54r) and fitted to the TF calibration curve. The TF grafting yield on liposomes was calculated from the calibrated TF amount relative to the 100% input.

### 4.11. Stability Study

The stability of liposomes and TF–liposomes was evaluated after storage at 4 °C for 14 days. Particle size was determined by ZetaSizer Nano with DTS1070 folded capillary zeta cell. Results were then processed using one-way Analysis of Variance (ANOVA). Ordinary one-way ANOVA followed by Tukey’s post hoc test was used to compare size differences between samples at different timepoints (Day 7 and Day 14, respectively). In addition to storage stability, the particle size and ζ-potential of liposomes and TF–liposomes were evaluated during incubation at 37 °C in PBS:FBS (1:1, *v*/*v*) for 24 h to assess colloidal stability in a physiologically relevant environment.

### 4.12. Microfluidic Device Making

Soft lithography techniques were applied to fabricate microfluidic devices made from polydimethylsiloxane (PDMS, Sylgard 184 Silicone Elastomer Kit Dow, Dow, Cheadle, UK), following approaches similar to those described in previous studies [65,66,67]. An acrylic mould (Weatherall Equipment and Instruments, Wendover, UK), 1000 µm thick, was laser-cut based on a design (see Appendix A Figure A3A) created in AutoCAD (version 2025.0.1, Autodesk, San Francisco, CA, USA) and subsequently affixed to an acrylic base. PDMS was prepared by combining the base elastomer with its curing agent in a 10:1 ratio, following the manufacturer’s guidelines. The resulting mixture was degassed for 30 min in a desiccator to eliminate trapped air. After degassing, the PDMS was poured into the moulds to a depth of 8 mm and cured in an oven at 80 °C for 1.5 h. Once cured, the PDMS structures were extracted from the moulds, and 2 mm and 1.5 mm diameter inlet and outlet ports were created for the medium and hydrogel channels, respectively, using a biopsy puncher. The PDMS components were then soaked in deionized water and sterilized via autoclaving. After drying overnight at 80 °C, the PDMS chips were bonded to #1 glass coverslips using a Corona plasma treatment system (Elveflow, Paris, France). Before device seeding, hydrophobicity was restored by placing the microfluidic chips at 80 °C for at least 4 h.

### 4.13. Two-Dimensional Cell Culture of BMECs, Astrocytes and Pericytes

Cells were maintained in a humidified incubator (37 °C, 5% CO_2_). Green fluorescent protein (GFP) labelled human brain microvascular cells (BMECs) (AngioProteomie; #cAP-0002GFP, Lot: 2022062302) were cultured in VascuLife vascular endothelial growth factor (VEGF) medium (Lifeline Cell Technology) supplemented with 10% FBS (Life Technologies). Cells were expanded in human fibronectin-coated (3 µg/cm^2^, Life Technologies) flasks, and media were changed every other day. Cells were passaged using TrypLE Express (Life Technologies) and used at passages 2–4 for experiments.

Primary human brain vascular pericytes (ScienCell; Cat#1200, Lot: 33155) and human brain astrocytes (ScienCell; #1800, Lot: 34727) were cultured as recommended by the manufacturer. Pericytes and astrocytes were expanded using pericyte medium (ScienCell, #1201) and astrocyte medium (ScienCell) in flasks coated with poly-L-lysine (2 μg/cm^2^, ScienCell). Cells were used at passages 2 or 3 for experiments and detached using 0.05% trypsin/EDTA (ScienCell).

### 4.14. BBB MVN Model Establishment

To establish the 3D microvascular BBB in vitro model, cells were incorporated into a fibrin hydrogel consisting of fibrinogen and thrombin, as described previously [38]. A 6 mg/mL bovine fibrinogen (Merck Life Science) was prepared in PBS (calcium/magnesium free, Life Technologies) by incubation for 2 h at 37 °C, followed by filtration through a 0.2 μm filter. Thrombin (Sigma-Aldrich) was initially prepared at a concentration of 100 U/mL by dissolving 1 KU thrombin powder in 10 mL PBS (calcium/magnesium free), followed by a 1:25 dilution in Vasculife VEGF endothelial medium prepared with 0.1875 U/mL of heparin and 2% FBS. Thrombin and fibrinogen solutions were prepared under sterile conditions and kept on ice during all procedures.

BMECs, astrocytes and pericytes were detached, centrifuged, and resuspended in culture media to be counted. Cells were spun down again separately and resuspended in thrombin and then combined. The thrombin–cell mixture was mixed in equal parts with fibrinogen to form the fibrin hydrogel. Cells were seeded within the fibrin hydrogel at a final concentration of 20 million cells/mL, 1.5 million cells/mL and 3 million cells/mL for BMECs, pericytes and astrocytes, respectively. The fibrin–cell mixture was immediately injected into the central channel of the microfluidic device. Seeded devices were placed in a sterile humidified Petri dish, and fibrin was allowed to polymerize for 30 min within an incubator (37 °C, 5% CO_2_). Finally, Vasculife VEGF endothelial medium prepared with 0.1875 U/mL of heparin and 2% FBS was added to the media channels. Media were changed every day in the devices.

To prevent liposomes from freely diffusing into the hydrogel without entering the MVN first, a monolayer of BMECs was seeded along the gel interface at day 5 of device culture. To improve cell attachment, media channels were emptied and coated with human fibronectin diluted in culture media (0.03 mg/mL, Life Technologies) for 10 min. BMECs were detached and seeded into one media channel at a concentration of 1.5 million cells/mL in device culture media. The device was tilted 45° for 10 min to improve monolayer formation along the gel via gravity. Subsequently, the same was performed for the opposing media channel.

### 4.15. Three-Dimensional Immunofluorescence Staining and Imaging

At day 7 of BBB MVN culture, immunofluorescence staining was performed in a similar fashion as described previously [38]. Briefly, media were removed, and devices were washed and then fixed with 4% paraformaldehyde (PFA; Electron Microscopy Sciences) overnight at room temperature (RT). Subsequently, devices were washed 3 times for 5 min with PBS at RT. Cell membranes were permeabilized with 0.1% Triton X-100 (Sigma-Aldrich) in PBS for 20 min at RT, followed by blocking in 0.1% normal goat serum (Abcam) prepared in 0.1% Triton X-100 for 1.5 h at RT. Immunostaining was performed using a primary antibody against glial fibrillary acidic protein (GFAP; #ab33922, Abcam) diluted 1:100 in the blocking buffer and incubated for 24 h at 4 °C. Devices were washed 3 times for 5 min. Subsequently, secondary antibodies (anti-rabbit Alexa Fluor 647; #4414, Cell Signalling) were diluted in PBS and applied for 2 h at RT. For visualization of filamentous actin, samples were incubated with phalloidin probes (165 nmol; #A12381, Invitrogen) for 20 min at RT. Nuclear staining was performed with DAPI (1:1000; Sigma-Aldrich) for 15 min. Finally, samples were washed again with PBS (3 times, 5 min each at RT) before imaging on an inverted confocal microscope (BC43, Olympus, Tokyo, Japan).

### 4.16. Liposome Perfusion Across the BBB MVNs

Six days after BBB microfluidic devices were seeded, TF-functionalized liposomes and non-functionalized liposomes were perfused through the MVNs at a concentration of 30 μg/mL. All liposomes were freshly prepared before each experiment and vortexed for 10 s before their use. For liposome perfusion, all media were removed, and on one side, the filled media channel (~100 μL) was replaced with 80 μL. Then pipette tips, with the ends cut to fit tightly, were inserted into the openings of the media channels. On the side where no media were present, 250 μL of liposome solution was added to one tip. This setup allowed liposomes to flow across the media channel (similarly to arterial flow) but also into the MVNs (capillary flow) due to the hydrostatic pressure difference between the two media channels.

### 4.17. Liposome Permeability Measurements Using Automated Image Processing

Thirty minutes after liposomes were perfused into the microvasculature, z-stacks of a region of interest (ROI) (1223 μm × 1583 μm) in the central part of the gel channel were taken by confocal microscopy (BC43, Olympus, Tokyo, Japan). Z-stacks covered the whole depth of the microvasculature. The same procedure was performed again 24 h later with the same imaging settings.

To quantify whether liposomes travelled outside the microvasculature, a FIJI [68] macro was applied to each of the z-stacks (Figure 4A) [43]. In summary, the binary mask of the vasculature was used to either quantify the signal intensity of liposomes within this outline or outside of this outline to obtain information about the mean fluorescent signal intensity inside or outside the vasculature, respectively [43]. Mean signal intensity for each z-slice was then averaged across the z-stack to obtain overall intensity values. Because the analysis was performed using an automated macro, all measurements were effectively blinded to experimental conditions. Raw measurements from TF–liposomes were normalized to the control measurement for each measurement day to prevent day-to-day variability in microscope imaging intensity, using Equation (2) below:(2)$$\Delta \;Cy3\;signal\;intensity=Cy3\;signal_{Transferrin\;Liposomes}-mean(Cy3\;signal_{Control\;Liposomes})$$

### 4.18. Liposome Permeability Assessment with Manual Signal Measurements up to 100 µm from the Vessel Lumen

To characterize the kinetics of Cy3-MAPT-ASO-loaded liposome transport across the microvasculature, fluorescence intensity profiles were quantified across the vessel lumen and the surrounding 100 µm of tissue, as shown in Appendix A Figure A5. Confocal z-stacks were first merged using average intensity projections in FIJI [68]. In a blinded manner, line scans were drawn perpendicular to selected vessel segments, and the fluorescence intensity of both GFP (microvasculature marker) and Cy3 (ASO signal) was measured using the ‘Line’ and ‘Plot profile’ functions in FIJI. The vessel boundaries were defined based on the GFP intensity profile, which was analyzed in Origin Pro (version 2024b, OriginLab Corporation, Northampton, MA, USA) using the ‘Level Crossing’ application to identify luminal borders (Figure 4C). The Cy3 fluorescence intensity was then plotted over a 100 µm distance extending outward from the vessel wall, normalized (%) to the mean luminal intensity, and corrected by subtracting background fluorescence. For each condition, at least 50 evenly distributed measurements were collected across three biological replicates and averaged to generate representative intensity profiles.

### 4.19. Liposome Toxicity Assessment via Microvasculature Network Analysis

Following the same workflow in Origin Pro as described in the previous section, vessel size was measured by determining the borders of the microvasculature via the intensity profile from the GFP expressed at the microvasculature. Vessel sizes were compared between different experimental conditions.

### 4.20. Statistics

All statistical analyses were performed using GraphPad Prism (version 10, GraphPad Software, Boston, MA, USA) or Origin Pro software. The ANOVA test is used to compare the differences between three or more groups of normally distributed samples. Results with a *p* value < 0.05 were considered significant. * *p* < 0.05; ** *p* < 0.01; *** *p* < 0.001; **** *p* < 0.0001. *n* indicates the number of independent experiments performed for the same procedure.

## 5. Conclusions

In conclusion, this study presents transferrin-functionalized liposomes that enhance the delivery of tau-targeting antisense oligonucleotides (MAPT-ASOs) across a physiologically relevant 3D in vitro model of the human BBB. The used MAPT-ASO effectively reduced tau expression and protected against axonal degeneration in neurons, confirming both delivery and functional efficacy. These findings suggest that the receptor-targeted liposomal system may enable systemic administration of ASOs for the treatment of tauopathies, potentially overcoming the limitations of current intrathecal approaches. This study also highlights the utility of the 3D BBB model as a translational platform for evaluating and optimizing brain delivery of nucleic acid-based therapeutics.

## Figures and Tables

**Figure 1 ijms-26-11347-f001:**
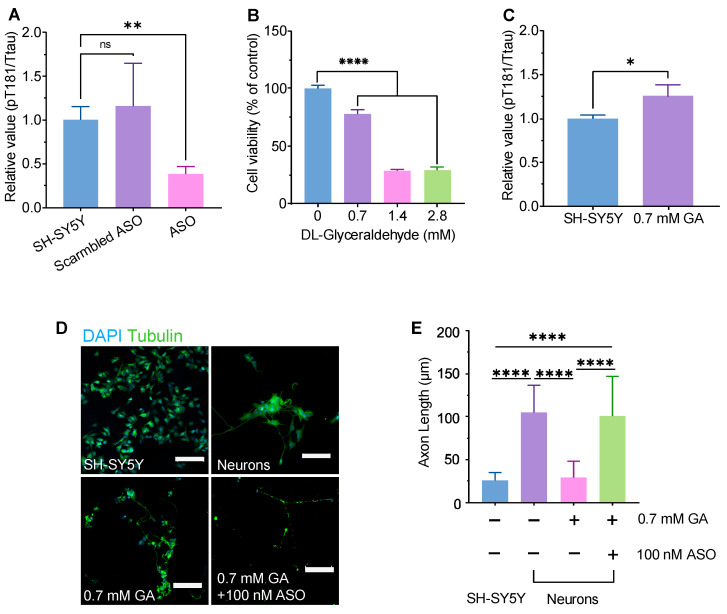
Assessment of MAPT-ASO effectiveness: (**A**) Quantification of pT181/tTau ratio in SH-SY5Y cells after 72 h treatment with vehicle, scrambled ASO, or MAPT-ASO. Data are mean ± SD, *n* = 3. (**B**) GA dose-dependently induced cell death in SH-SY5Y cells. Data are mean ± SD, *n* = 3. (**C**) Quantification of pT181/tTau ratio in SH-SY5Y cells after 72 h treatment with 0.7 mM GA. Data are mean ± SD, *n* = 3. (**D**) Confocal laser scanning microscopy (CLSM) images of axons (tubulin, green) and nuclei (DAPI, blue) in SH-SY5Y cells, RA-differentiated neurons, neurons treated with 0.7 mM of GA, and neurons cotreated with 0.7 mM of GA and 100 nM of MAPT-ASO. Scale bar, 100 μm. (**E**) Axon length quantification for the four conditions in (**D**). Data are mean ± SD, *n* = 3. Statistics applied to the figure: *p* values are calculated using one-way ANOVA with Dunnett’s multiple comparisons test vs. the relative control. Non-significant (ns) *p* > 0.05, * *p* < 0.05, ** *p* < 0.01, **** *p*  <  0.0001. Abbreviation: GA, glyceraldehyde.

**Figure 2 ijms-26-11347-f002:**
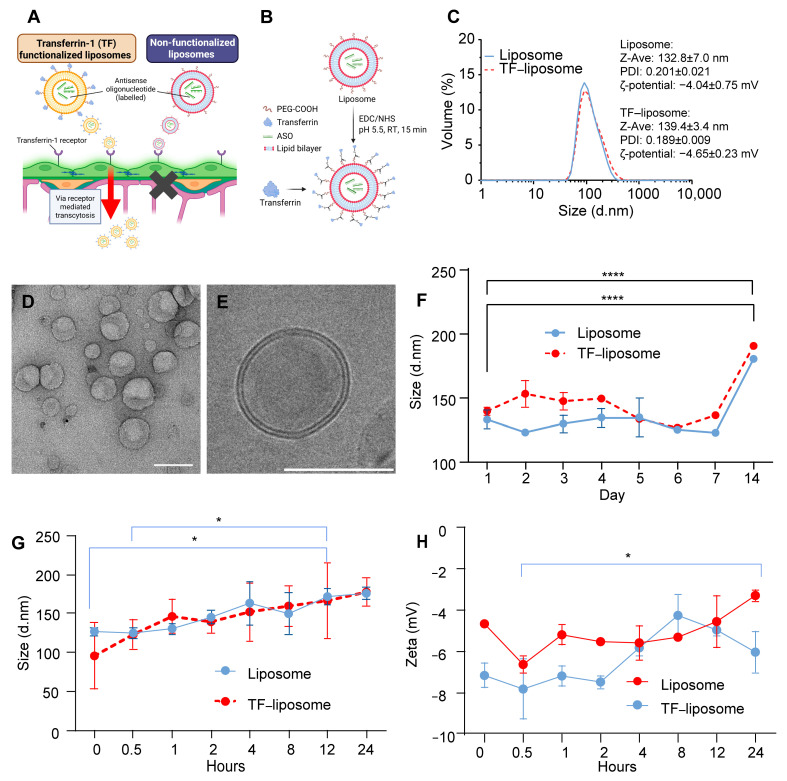
Characterization of ASO-loaded non-functionalized and transferrin-functionalized (TF) liposomes: (**A**) Schematic illustration of the mechanism of TF–liposomes travelling across the BBB via receptor-mediated transcytosis. The arrow signifies successful traversal of the BBB, while the cross indicates failure to cross it. Created in BioRender. Konig, S. (2025) https://BioRender.com/p8eyub0 (accessed on 18 November 2025). (**B**) Schematic illustration of transferrin conjugation onto liposomes. (**C**) Representative size distribution, PDI, and ζ-potential of liposome and TF–liposome preparations. (**D**) Morphology of liposomes shown in transmission electron micrograph (TEM) (**D**) and cryogenic-TEM (**E**) images. Scale bar, 100 nm. (**F**) Stability of liposomes and TF–liposomes over 14 days at 4 °C. (**G**) Hydrodynamic diameter of liposomes and TF–liposomes changes during incubation at 37 °C in PBS:FBS (1:1, *v*/*v*) for 24 h. (**H**) Zeta potential of liposomes and TF–liposomes measured during incubation at 37 °C in PBS:FBS. Data are mean ± SD, *n* = 3. Statistics applied to the figure: *p* values are calculated using one-way ANOVA with Dunnett’s multiple comparisons test vs. the relative control. * *p* < 0.05, **** *p* < 0.0001.

**Figure 3 ijms-26-11347-f003:**
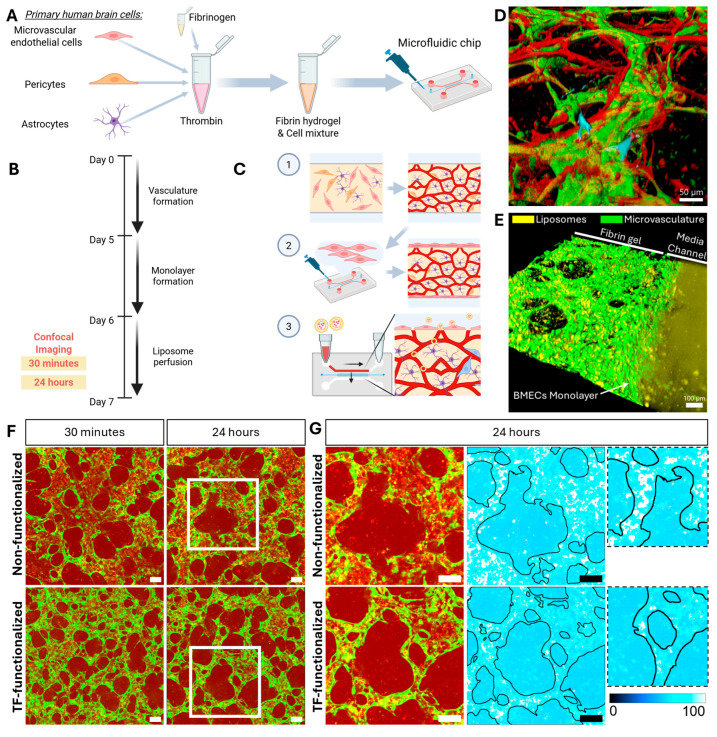
Liposome perfusion within a microvascular blood–brain barrier model: (**A**) Human brain cells are incorporated into a fibrin gel within a microfluidic chip (see Appendix A Figure A3 for the microfluidic device design). (**B**) Timeline of experiments. Confocal images of microvasculature and fluorescently labelled liposomes are taken 30 min and 24 h after liposome perfusion. (**C**) (1) After cells are incorporated into a hydrogel, a mature BBB microvasculature is formed. (2) To prevent liposomes from perfusing into the hydrogel, a monolayer of BMECs is formed at day 5 along the hydrogel–media interface. (3) At day 6, liposomes are perfused across and into the microvasculature using a hydrostatic pressure gradient. Created in BioRender. Konig, S. (2025) https://BioRender.com/2js7u8q (accessed on 18 November 2025). (**D**) Confocal image of the microvasculature formed by BMECs (green) with pericytes (red) and astrocytes (blue) aligned. (**E**) Confocal image of liposomes (yellow) entering the microvasculature (green). (**F**) Average intensity z-projected images of microvasculatures (green) perfused with liposome (red). (**G**) Magnified images (as indicated by white squares in panel F). First column shows both microvasculature (green) and liposomes (red). Middle panel and corresponding magnified insets showing outline of vasculature and intensity map of liposomes with high signal (white) within the lumen of the microvasculature for the non-functionalized liposomes (**upper row**) and TF-functionalized liposomes (**lower row**). Scale bars = 100 µm.

**Figure 4 ijms-26-11347-f004:**
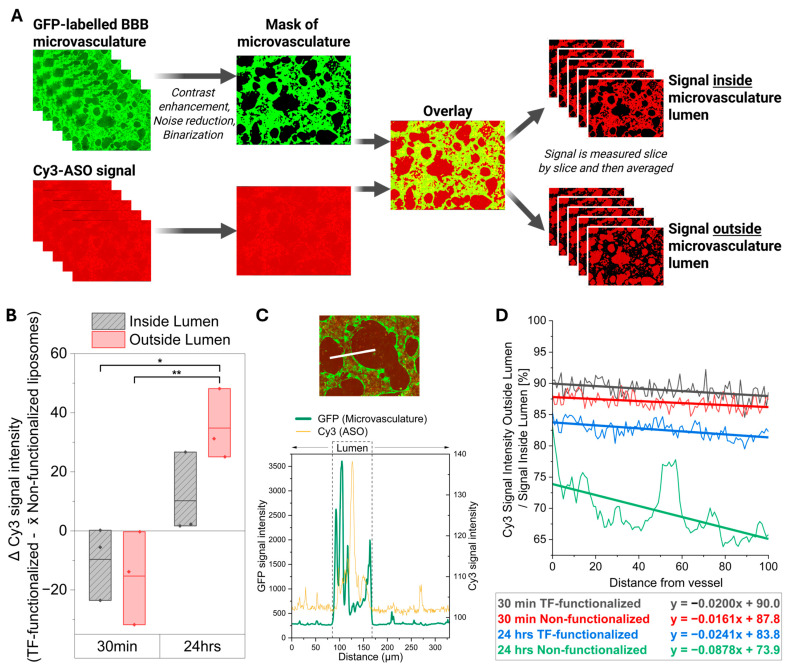
Quantifications of liposome-mediated delivery of Cy3-MAPT-ASO within a microvascular BBB model: (**A**) First method to evaluate the permeability of Cy3-MAPT-ASO. Binarized images of GFP-labelled microvasculature were used to segregate Cy3-MAPT-ASO signal within or outside the microvasculature lumens independently for each Z-plane [43]. (**B**) Results of quantifications presented as the difference (Δ) in measured Cy3 signal intensity, calculated by subtracting the mean (x̄) signal of non-functionalized liposomes from that of samples with TF-functionalized liposomes. *n* = 3 devices per experimental condition, from two independent experiments. Statistics applied to the figure: *p* values calculated using two-way ANOVA with Tukey’s multiple comparisons test. * *p* < 0.05; ** *p* < 0.01. (**C**) As a second method of liposome permeability assessment, Cy3-MAPT-ASO signal intensity manually measured across the microvascular lumen and its surrounding tissue. The upper panel shows an example microvascular lumen, and the lower panel displays the corresponding fluorescence intensity profile along the white line indicated in the upper image. (**D**) Averaged fluorescent signal intensity from > 50 measurements across three biological replicates plotted as a function of distance from the lumen, normalized to intensity within the lumens. Linear fits are applied, and the resulting equations are displayed. Measurements are taken in three devices per condition across two independent experiments.

## Data Availability

Dataset available on request from the authors. The analysis code is available at https://github.com/S-Konig/Konig_Fluo_Signal_Quant_BBB_Microvascular_networks (accessed on 18 November 2025). A digital object identifier accession for code is available at https://doi.org/10.5281/zenodo.17047899 (Ref. [43]).

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
