# Peer review of "Transferrin-Functionalized Liposomes Enhance MAPT-ASO Transport Across a 3D Blood–Brain Barrier Microvascular Network Model"

_ijms, 2025, doi:10.3390/ijms262311347_

Round 1
Reviewer 1 Report
Comments and Suggestions for Authors
The manuscript presents an interesting and well-structured study exploring the use of transferrin-functionalized liposomes to enhance the transport of antisense oligonucleotides (MAPT-ASO) across a 3D blood–brain barrier (BBB) microvascular model. The topic is highly relevant, and the experimental platform employed shows strong potential for studying brain delivery of nucleic acid–based therapeutics.
However, several aspects could be further investigated or discussed to strengthen the manuscript:
Liposome characterization:
The physicochemical characterization (size, PDI, ζ-potential, morphology) is appropriate but could be expanded with additional controls such as quantification or confirmation of transferrin conjugation stability. Considering also data on batch-to-batch reproducibility would also improve the robustness of the formulation, is available.
Stability under different conditions:
Stability was only assessed for a short period and at 4 °C. It would be valuable to include or at least discuss experiments assessing stability under physiological or storage-relevant conditions (e.g., at 37 °C, in serum, or in simulated plasma) to evaluate the system’s robustness in biologically relevant environments.
Interaction with biological fluids:
Since systemic administration is the intended route, the potential formation of a protein corona and its impact on transferrin receptor affinity and biodistribution should be considered. A discussion of this aspect, supported by relevant literature, would provide useful context for in vivo translation.
Potential for BBB crossing in vivo:
While the 3D BBB model is sophisticated and informative, a brief discussion on how the observed results might translate in vivo would strengthen the paper. For instance, possible strategies to evaluate BBB crossing and ASO delivery in animal models, or comparison with known transferrin-targeted systems, could be outlined.
Reviewer 2 Report
Comments and Suggestions for Authors
This manuscript, entitled “Transferrin-Functionalized Liposomes Enhance MAPT-ASO Transport Across a 3D Blood–Brain Barrier Microvascular Network Model,” has been carefully reviewed. The following comments and suggestions are provided to help the authors improve the clarity, rigor, and overall impact of their study.
The novelty of this study should be clearly highlighted (In abstract, introduction, and discussion).
In the Materials and Methods section, it is recommended to include a complete list of all materials used, with their sources and specifications, in a separate subsection titled “Materials.”
Regarding the ASO dose, the manuscript mentions the use of 1.35 mM of both scrambled and target ASO. A more detailed justification of how this concentration was selected whether based on prior studies or preliminary optimization would help readers better understand the rationale for this dosage.
Additionally, please review line 397 (“MEM”) for clarity or correction.
It would be useful to add a short subsection describing the study design and clearly defining the experimental groups and treatment conditions.
For greater reproducibility, the authors should provide a more detailed explanation of how the percentage of penetration up to 100 μm from the lumen and the criteria for ROI selection were calculated.
While the 24-hour results convincingly demonstrate the superiority of TF-functionalized liposomes, the differences at the initial 30 minute time point were relatively small. The authors could further elaborate on the mechanistic explanation for this gradual increase in permeability whether it depends on receptor density, internalization kinetics, or other contributing factors.
Evaluation of this delivery system in relevant in vivo models would strengthen the translational significance of the findings, since physiological blood flow and BBB dynamics in vivo may influence ASO distribution.
The manuscript also mentions slight liposome aggregation during microvascular perfusion; potential strategies for improving colloidal stability under flow conditions (e.g., altering PEG chain length, lipid composition, or TF surface density) could be briefly discussed.
The conclusion could be strengthened by more explicitly summarizing the key findings and potential therapeutic benefits of this delivery strategy.
The discussion should more clearly emphasize the novelty of the work, especially how integrating TF-functionalized liposomes with a 3D BBB microvascular platform advances current CNS drug delivery methodologies.
It is also recommended that the authors carefully review all abbreviations, gene names, and protein symbols throughout the manuscript to ensure consistency and accuracy. Finally, the statistical analyses should be described in more detail, including specific information on tests used, sample sizes, and significance levels (p-values), and corresponding results should be clearly reported in both the figures and text to enhance transparency and scientific rigor.
Round 2
Reviewer 2 Report
Comments and Suggestions for Authors
Dear authors,
I want to express my sincere appreciation for your efforts in addressing the reviewers' comments and implementing the necessary revisions in your manuscript. The changes you have made have significantly improved the scientific quality and precision of the research, demonstrating your attention to detail and commitment to enhancing the overall quality of the study.
Your work provides valuable insights into the effects of transferrin-functionalized liposomes on MAPT-ASO transport across 3D blood-brain barrier models, and it will undoubtedly contribute to further advancements in this field.
I wish you continued success with your research and hope your manuscript will be successfully published.
Best Regards